# The Bovine Tuberculoid Granuloma

**DOI:** 10.3390/pathogens11010061

**Published:** 2022-01-04

**Authors:** Mitchell V. Palmer, Carly Kanipe, Paola M. Boggiatto

**Affiliations:** Bacterial Diseases of Livestock Research Unit, National Animal Disease Center, Agricultural Research Service, United States Department of Agriculture, Ames, IA 50010, USA; carly.kanipe@usda.gov (C.K.); paola.boggiatto@usda.gov (P.M.B.)

**Keywords:** bovine, granuloma, lung, lymph node, mycobacteria

## Abstract

The bovine tuberculoid granuloma is the hallmark lesion of bovine tuberculosis (bTB) due to *Mycobacterium bovis* infection. The pathogenesis of bTB, and thereby the process of bovine tuberculoid granuloma development, involves the recruitment, activation, and maintenance of cells under the influence of antigen, cytokines and chemokines in affected lungs and regional lymph nodes. The granuloma is key to successful control of bTB by preventing pathogen dissemination through containment by cellular and fibrotic layers. Paradoxically, however, it may also provide a niche for bacterial replication. The morphologic and cellular characteristics of granulomas have been used to gauge disease severity in bTB pathogenesis and vaccine efficacy studies. As such, it is critical to understand the complex mechanisms behind granuloma initiation, development, and maintenance.

## 1. Introduction

The granuloma is a distinctive morphological lesion associated with chronic antigenic stimulation. It is considered the hallmark lesion of tuberculosis but can also be seen in response to other persistent stimuli of bacterial, fungal, protozoal or foreign body origin [1,2,3]. First described in 1679, it was termed a *tubercle*, an appellation that would be used by Robert Koch when he described *tubercle bacilli*, over 200 years later [4]. A granuloma may be referred to as a *tuberculoid granuloma*, when the etiology is a member of the *Mycobacterium tuberculosis* complex, which includes *M. tuberculosis*, *M. bovis*, *M. africanum*, *M. microti*, *M. caprae*, *M. canetii*, *M. pinnipedii*, *M. orygis*, *M. suricattae*, *M. mungi*, the dassie bacillus, and the chimpanzee bacillus [5,6,7,8,9,10,11].

Despite the importance of the granuloma in the bovine immune response to *M. bovis*, little is understood about the immune response dynamics at the granuloma level [12]. The pathogenesis of bovine tuberculosis (bTB), and thereby the process of bovine tuberculoid granuloma development, involves the recruitment, activation, and maintenance of cells under the influence of antigen, cytokines and chemokines in affected lungs and regional lymph nodes. When infection occurs via aerosol exposure, some bacilli pass beyond the upper respiratory tract and are deposited in terminal bronchioles and alveoli where they are phagocytosed by resident alveolar macrophages. The “droplet nuclei” theory of aerosol infection described for human tuberculosis likely holds true for aerosol exposure in cattle. Tiny residues, <5 μm, of evaporated droplets (droplet nuclei) are generated by coughing or heavy breathing [13,14,15]. Droplet nuclei become infectious when they contain or are associated with one to several bacilli expelled by an infected host. Larger droplets settle out within a short distance from their source, while droplet nuclei can remain airborne for hours [13,14]. Once inhaled, infectious droplet nuclei reach terminal bronchioles and pulmonary alveoli, while larger droplets >5 μm are removed in the upper respiratory tract [13,14,16] via the mucociliary apparatus, which transports bacilli upwards toward the pharynx, where the material may be swallowed or expelled by coughing. 

Alveolar macrophages are the first immune cell to encounter inhaled bacilli. They are a key component of the innate immune defenses of the lung and function to limit bacterial replication through phagocytosis of bacilli, phagolysosome fusion, phagolysosome acidification, lysosomal proteolytic enzymes, and production of antimicrobial reactive oxygen and nitrogen species. As specialized intracellular pathogens, virulent mycobacteria can manipulate host macrophages such that they are able to reside and persist within them, rather than be destroyed [17]. Virulent mycobacteria resist killing through strategies that inhibit phagolysosome fusion and acidification, effectively arresting phagosome maturation [18,19,20]. Surviving bacilli establish intracellular residence within these arrested phagosomes. Virulent mycobacteria species use proteins such as early secreted antigenic target (ESAT)–6 to lyse phagosomal membranes [21], gaining entry into the macrophage cytosol. Within the cytosol, mycobacteria multiply, eventually causing macrophage cell death, which prompts the release of chemokines and cytokines recruiting other cells to the site of infection, thereby perpetuating the inflammatory cycle. 

As part of the adaptive immune response, some macrophages and dendritic cells containing bacilli exit the alveolar spaces and enter the pulmonary interstitium where they access lymphatic or hematogenous systems and spread to local lymph nodes (tracheobronchial and mediastinal). There, through antigen presentation, naïve T cells recognize their cognate antigens, proliferate, and gain effector functions; thus, initiating an adaptive immune response against mycobacteria. From the lymph nodes, primed, antigen-specific T cells migrate to the site of infection and further activate monocytes and macrophages to limit or eliminate mycobacterial replication. This, in addition to recruitment of macrophages and lymphocytes contributes to the developing granuloma. Paradoxically, while the granuloma is considered a host-protective structure by controlling and containing infection, the granuloma also provides a niche for early bacillary proliferation. The host response also leads to tissue destruction and necrosis. 

The associated lesions of the lung and pulmonary lymph nodes are known as the primary complex or Ghon complex (named for Anton Ghon (1866–1936)) [22]. In cattle, the primary complex is easily overlooked as the initial lung lesions may be very small (<1 cm); most (70%) are only found through careful slicing of the lung [23]. Although activities within the lung and associated lymph nodes are coordinated, they differ in their kinetics, cellular composition, and cellular trafficking. It is, therefore, not surprising that the cytokine environment in developing granulomas of the lung differ from those of the associated lymph nodes, even when examined at the same time and in the same animal [24]. For example, it has been shown, in cattle experimentally infected via aerosolized *M. bovis*, that late-stage granulomas in the lung showed greater expression of interferon gamma (IFN-γ), transforming growth factor beta (TGF-β), interleukin (IL)-10 and IL-22 mRNA than similar-late stage granulomas in the associated tracheobronchial lymph node. In contrast, expression of IL-17A mRNA was greater in granulomas of the tracheobronchial lymph node compared to the lung [24].

## 2. Granuloma Staging

It is generally accepted that granulomas are developmentally dynamic, changing over time. Categorization of bovine tuberculoid granulomas using a system put forward by Wangoo [25] has become common, if not standard for bTB researchers (Figure 1). 

In this system, microscopic morphologic characteristics including size, cellular composition, and presence or absence of necrosis and fibrosis are used to categorize granulomas into four (I-IV) separate, but slightly overlapping stages. Stage I granulomas contain infiltrates of epithelioid macrophages with low numbers of lymphocytes, granulocytes, and multinucleated giant cells (MGCs). Acid fast bacilli (AFB) are not numerous, but when present are seen intracellularly in macrophages and MGCs. Importantly, in stage I, necrosis is absent. Stage II granulomas are characterized by nodular to irregular accumulations of epithelioid macrophages, with granulocytes, lymphocytes and MGCs also present. The presence of variable amounts of central necrosis sets stage II granulomas apart from stage I. Acid-fast bacilli, when present are seen in macrophages and MGCs, but are also seen free within the necrotic caseum. Stage III granulomas are characterized by central necrotic cores surrounded by a zone of epithelioid macrophages admixed with MGCs and lymphocytes. Lymphocytes become more numerous and macrophages less numerous as distance from the necrotic core increases. Frequently, portions of the necrotic core are mineralized through a process known as dystrophic mineralization. Mineralized foci are often surrounded by MGCs. In this stage, AFB may be seen in moderate numbers and are most often located extracellularly, free within the necrotic caseum, although they can also be seen within MGCs, and less commonly, in macrophages. Encapsulation of the necrotic core and cellular infiltrate by a complete fibrous capsule distinguishes this stage from earlier stages. The final stage, stage IV represents a coalescence of stage III granulomas with multiple necrotic and partially mineralized centers, surrounded and subdivided by thick fibrous bands. Acid-fast bacilli may be numerous and are most seen free within the necrotic caseum, often in areas of mineralization.

This four-stage model has allowed bTB investigators to communicate using a common reference system and has proven particularly useful in evaluation of bTB vaccine candidates. The vaccine most investigated for bTB is the human vaccine *M. bovis* bacillus Calmette-Guérin (BCG). In cattle, BCG does not always prevent infection or lesion development; however, it consistently decreases lesion severity at the gross and microscopic level. Several BCG vaccine trials in cattle have documented a decrease in the number of stage III and IV granulomas in vaccinates [26,27,28,29,30,31]. These late-stage granulomas are associated with extensive necrosis, increased tissue destruction and increased bacterial burdens, and as such, it has been hypothesized that with fewer late-stage granulomas, there would be less bacterial shedding and decreased disease transmission. Consequently, although BCG may not provide sterile immunity, it should result in decreased transmission and decreased incidence within a given population.

Most studies in cattle have examined tuberculoid granuloma development at later time points, typically 90 days after infection and beyond [24,25,26,32,33,34]. Granulomas of unknown duration from naturally infected cattle have also been described in detail [35,36]. At ≥90 days after infection, microscopic examination reveals the presence of all 4 granuloma stages. In fact, granulomas of two or more stages are often present on the same microscopic section [37]. A relatively common finding is the presence of stage I granulomas, separate, but near stage III-IV granulomas. Stage I granulomas such as these have been categorized as “satellite granulomas” (Figure 2) [26,33]. 

Although morphologically similar in terms of cellular composition and granuloma stage, it is not clear if stage I granulomas of early infection are functionally similar to stage I satellite granulomas, which develop later in the disease process. The finding of more satellite granulomas in non-vaccinated cattle compared to BCG-vaccinated cattle and the lower number of AFB in satellite stage I granulomas compared to early disease stage I granulomas suggest that there may be functional differences [26,33].

## 3. Early Lesion Development

A less investigated area of bTB pathogenesis has been very early granuloma development (i.e., pre-stage I). In very early lesions, such as those seen 15 days after aerosol infection, the 4-stage model described above is less useful. In this time frame, the nomenclature used for describing early lesions in rhesus macaques experimentally infected with *M. tuberculosis* is more appropriate [38]. In this typing scheme, type 1 lesions are considered pre-granulomas; unorganized, lacking defined boundaries or peripheral lymphocyte-rich zones and composed of small foci of macrophages and few lymphocytes that expand alveolar septa and extend into the alveoli.

Type 2 lesions are composed of similar inflammatory cells as type 1 lesions, still lacking a peripheral lymphocyte-rich zone, but are more circumscribed and roughly circular with semi-demarcated borders, as compared to type 1 lesions. Type 2 lesions are most similar to stage I lesions from the above four-stage cattle model. Type 3 lesions are similar to type 2 lesions but contain small areas of necrosis, characterized by loss of cellular detail, nuclear pyknosis (condensed nuclei), and karyorrhexis (fragmented nuclei). These are most similar to stage II lesions in the cattle model. Type 4 lesions are organized, well-circumscribed granulomas consisting primarily of macrophages admixed with lesser numbers of granulocytes and variable numbers of peripheral lymphocytes (non-necrotizing granulomas). Type 5 lesions are similar to type 4 lesions but exhibit central necrosis (necrotizing granulomas). Types 4 and 5 are similar but not identical to stage III granulomas from the above cattle model. The pronounced fibrotic response characteristic of bovine tuberculoid granuloma stages III and IV are not characteristic of early lesions in macaques, and therefore, not described in the macaque typing scheme. 

The early, pre-stage I phases of pulmonary granuloma development in cattle have been challenging to examine. The large volume of lung tissue, even in a calf, makes identification of very small lesions difficult. Even so, lesions from cattle at 7–15 days post-infection have been described [38,39]. Gross lesions are small (≤1 mm), dark red, punctate foci. At this early time point, microscopic organized granulomas are not evident, but rather the interstitium is expanded due to congestion and increased numbers of lymphocytes, macrophages and low numbers of granulocytes, often present around blood and lymphatic vessels [38]. Proteinaceous fluid and fibrin strands variably fill alveoli, likely due to increased vascular permeability. Expression of chemokines such as monocyte chemoattractant protein-1 (MCP-1) and chemokine (C-X-C motif) ligand 9 (CXCL9) attract CD68^+^ macrophages (Figure 3), some of which contain a large amount of highly vacuolated cytoplasm, consistent with descriptions of foamy macrophages [38]. MCP-1 is thought to be primarily responsible for recruiting monocytes, dendritic cells, natural killer cells and activated T cells, while CXCL9, under the influence of IFN-γ, is a macrophage and T cell chemoattractant, which also promotes extravasation of cells from the blood to the tissue. It is therefore appropriate that these potent chemokines have been shown to be expressed early as infiltrates of macrophages and T cells form the developing bovine tuberculoid granuloma [38,39].

## 4. Epithelioid Macrophages, Foamy Macrophages and Multinucleated Giant Cells

Macrophages are seen at all granuloma stages; their number, however, decreases as development progresses. In stages I and II, a high percentage of the cells within the granuloma are macrophages, whereas in stages III and IV, macrophages generally form a small rim around the necrotic core, with fewer in the outermost layers of the granuloma [40]. Although not yet described in cattle, other models of tuberculosis have shown bacilli-infected macrophages entering and leaving granulomas, likely an important means of disease dissemination [41].

Within the granuloma there is constant tension between proinflammatory forces aimed at pathogen elimination, and forces to maintain tissue integrity and initiate tissue repair. Through, as yet uncharacterized signals from virulent mycobacteria, polarization of the macrophage population in to classically activated M1 (effector functions aimed at pathogen removal) and alternatively activated M2 (function aimed at eliminating dead or damaged cells and matrices combined with tissue repair) macrophages may occur [42]. Although M1- and M2-type macrophages have not been fully characterized in bovine tuberculoid granulomas, there are macrophage populations with pro- and anti-inflammatory phenotypes, which are spatially distributed within the granuloma; macrophages near the necrotic core being pro-inflammatory and those more peripherally located being anti-inflammatory [43]. In mice, expression of inducible nitric oxide synthase (iNOS) is a hallmark of proinflammatory M1 macrophages, while anti-inflammatory, healing M2 macrophages express arginase 1 [44]. iNOS immunoreactivity has been demonstrated by immunohistochemistry (IHC) in macrophages and MGCs in bovine tuberculoid granulomas [45]. Macrophage activation is a dynamic process dependent on the presence and persistence of both host-derived and pathogen-derived stimuli, therefore, at any given time within a granuloma, there is a heterogenous population of macrophage phenotypes. 

Macrophages within tuberculoid granulomas can transdifferentiate into various types, including epithelioid macrophages, foam cells, and MGCs [4,46]. Epithelioid macrophages are a hallmark of granuloma formation. In traditional hematoxylin and eosin (H&E) stained microscopic sections, epithelioid macrophages have a pale pink granular cytoplasm with indistinct cell borders. The large, elongated, ovoid nucleus is less dense than that of a lymphocyte, and surrounded by abundant cytoplasm filled with lysosomes, mitochondria, and a prominent Golgi apparatus [47,48]. Uniquely, epithelioid macrophages are joined together through interdigitating membranes and tight junctions (i.e., adherens junctions, tight junctions and desmosomes), typical characteristics of epithelial cells rather than traditional macrophages [2,49,50]. Additionally, like epithelial cells they express E-cadherin, α-catenin, plakoglobin and zonula occludens-1 proteins, all protein components of cell-to-cell adhesion molecules [49]. In terms of function, epithelioid macrophages are believed to be less phagocytic and more secretory than activated macrophages [51,52]. In human mycobacteria-induced granulomas, the sources of the potent free radical nitric oxide (NO) are epithelioid macrophages and MGCs [53]. Studies to examine epithelioid macrophage transformation and the expression of epithelial cell-like characteristics remain to be conducted in bovine tuberculoid granulomas. 

Descriptions of bovine tuberculoid granulomas have infrequently included the mention of foamy macrophages [38], a subtype of macrophages, believed to be the result of mycobacteria-driven imbalances between the influx and efflux of low-density lipoproteins (LDL) molecules that contain cholesterol, triacylglycerides and phospholipids [54]. Ultrastructurally, within foamy macrophages, bacilli can be seen in close apposition to lipid droplets, which may serve as a nutrient source for bacilli. In humans, post-primary or reactivation tuberculosis begins with an accumulation of foamy macrophages within alveoli, usually in the upper lobes of the lung [55]. Degenerating foamy macrophages leave behind lipid-rich debris resulting in a lipid pneumonia, which then develops into foci of caseous necrosis. Post-primary tuberculosis has not been a feature of bTB, and foamy macrophages, although mentioned in some microscopic descriptions of bTB, have not been well characterized and their role in bTB pathogenesis remains unelucidated. 

Multinucleated giant cells have been noted in all studies of bovine tuberculoid granulomas. The role of MGCs during granuloma formation is incompletely understood. These unique cells form in environments of persistent antigen-induced inflammation. The requirement for antigen persistence is exemplified by the fact that MGC formation within a granuloma is a feature of persistent virulent, but not avirulent mycobacteria [56], possibly due to species differences in secreted products and cell wall components [57]. The fundamental mechanisms of MGC formation are not fully understood. It is known that MGC formation is a macrophage-specific response to persistent antigenic stimulus [2] and formation is mediated in part by IFN-γ and SIRPα (macrophage fusion receptor) [58,59]. Once believed to be the result of macrophage fusion [60,61], it has been suggested, that NO-induced DNA damage results in mitotic defects and multinucleation within individual macrophages [62,63]. Although believed to be poorly phagocytic, many MGCs in bovine tuberculoid granulomas contain one to several AFB (Figure 4) [56]. 

In cattle and other ruminants, the number of MGCs has been used as a measure of lesion severity, with higher numbers of MGCs associated with greater antigen persistence, increased inflammation, and more severe disease [35,64]. 

MGCs vary in nuclear organization and are categorized as either foreign body or Langhans type, having haphazardly arranged nuclei, or nuclei arranged in an arcuate, semicircular pattern, respectively. Both types of MGCs may be seen in bovine tuberculoid granulomas, however, Langhans MGCs are much more common [65]. The reason(s) for this are unknown and it is also unknown if there are functional differences between these two types of MGC. Using in situ hybridization (ISH), MGCs in bovine tuberculoid pulmonary granulomas have been shown to express TNF-α, IFN-γ, TGF-β, IL-17A and IL-10 [65]. Moreover, using ISH, MGCs of early stage I granulomas were shown to express more IL-17A and IL-10, and less TGF-β than MGCs of late-stage IV granulomas [65]. Using IHC, TGF-β, TNF-α, CD68, and iNOS expression have also been documented in MGCs in bovine tuberculoid lymph node granulomas [25,32,33].

## 5. Lymphocyte Response

As granulomas progress from stages I-IV an initial collection of CD68^+^ macrophages and CD68^+^ MGCs becomes a mixture of macrophages, MGCs and lymphocytes [25,26,32,33]. It is widely accepted that protective immunity to mycobacterial agents depends on interactions between macrophages and T cells [66]. As such, studies of cell types within bovine tuberculoid granulomas have described numerous CD3^+^ T cells [27] with CD4^+^ T cells as the predominant subtype, regardless of granuloma stage [33]. CD4^+^ T cells are a critical source of IFN-γ, which activates macrophages, thus facilitating killing of intracellular bacilli. Nevertheless, one study demonstrated a preponderance of CD8^+^ T cells, with 70% of stages III/IV having a CD4:CD8 ratio of <1.0 [37]. This same study described CD8^+^ T cells at the outermost boundaries of the lymphocyte rich zone of stage I/II granulomas [37]. It has been inferred from this spatial arrangement that CD8^+^ T cells play a role in the initial attempt to restrain granuloma growth. As infected macrophages are lysed by CD8^+^ T cells, activated, infiltrating macrophages with better bactericidal potential help limit bacterial replication [67,68].

Gamma delta (γδ) T cells are a subset of CD3^+^ T cells with functional characteristics of both the innate and adaptive immune system [12,69]. Particularly relevant to tuberculosis, they make up a significant portion of the immune cells of the respiratory tract mucosa. As part of their innate immune functions, they are often associated with the responses to invading pathogens. As such, most studies of bovine tuberculoid granulomas document their presence in early stage granulomas, decreasing in later stages [33,38], although some studies have found their numbers higher in late stage III/IV granulomas [25,32]. They have been documented in developing granulomas as early as 7–15 days after experimental infection [38,70] and are believed to promote containment of bacilli through cytokine and chemokine release and macrophage activation. Accordingly, they have been shown to produce IFN-γ at levels similar to CD4^+^ T cells, as well as CCL2, IL-10, IL-22, TGF-β and IL-17A [12,40,69,71,72,73].

The role of antibodies in granuloma development or bTB pathogenesis is unclear. However, CD79^+^ B cells have been identified by IHC scattered among macrophages and CD3^+^ T cells in granulomas [27]. In stages I/II, B cells are scattered among macrophages and are fewer in number than T cells. In more advanced granulomas, B cells are found peripherally, sometimes clustered subjacent to or within the fibrous capsule [25,70].

## 6. Neutrophils

Neutrophils are part of the innate immune response and are regarded as the first line of defense to invading pathogens. Neutrophils are short-lived, terminally differentiated, effector cells with three main antimicrobial functions: phagocytosis, degranulation and release of material in the form of neutrophil extracellular traps (NETs) [74,75] Upon arrival to the site of infection, neutrophils may phagocytose invading pathogens, including mycobacteria through Fc and complement receptors. Complex signaling pathways promoted by the phagocytic process, lead to the fusion of protease containing granules with the phagosome, triggering the oxidative burst [76]. Neutrophils may also discharge granules extracellularly, in effect killing extracellular bacteria, but also producing tissue damage [77]. Neutrophil extracellular traps are composed of a web of chromatin, histones and anti-bacterial proteins released from neutrophil granules. Transcriptionally active, neutrophils produce cytokines, modulate the activities of neighboring cells, contribute to the resolution of inflammation and have a role in innate immune memory (reviewed in [75]). 

In humans, neutrophils have been noted in advancing cavitary lesions, as well as nascent granulomas [78]. Likewise, neutrophils have been described in bovine pulmonary tuberculoid granulomas of all stages [25,79], although there has been more emphasis on their presence in early stages I/II [80]. In vitro studies have demonstrated the ability of bovine neutrophils to release NETs effective at killing *M. bovis* BCG. Interestingly, bovine neutrophils released more NET components in response to killed BCG than live BCG [74]. Neutrophils may be activated by many stimuli, including pathogen-associated molecular pattern (PAMPs), such as lipopolysaccharide, peptidoglycan, lipoteichoic acids, double stranded RNA and bacterial DNA, which are recognized by neutrophil pattern recognition receptors (PRRs). Neutrophil activation may lead to effective bacterial killing, but the neutrophil actions that lead to bacterial death may also produce tissue damage [81].

In naturally infected cattle, a positive correlation between neutrophil numbers and bacterial burden was seen, and it was suggested that higher bacterial loads result in persistent neutrophil recruitment, even in chronic, encapsulated granulomas [35]. Neutrophils may play an important role, particularly in early granuloma formation and are associated with the earliest signs of necrosis [39]. The first appearance of necrosis, seen 14–28 days after infection, is often associated with centrally located neutrophilic infiltrates [33,34]. One study noted that by day 14, clusters of neutrophils were surrounded by a mantle of macrophages [39]. Macrophages contained phagocytosed neutrophilic debris, a process known as efferocytosis, where macrophages phagocytose apoptotic neutrophils [77]. Other remnants of dead neutrophils become part of the necrotic caseum [82]. 

## 7. Cytokine Response and Bacterial Burdens

Although the T helper (T_H_)1/T_H_2 T cell paradigm described for mice, may not transfer precisely to bovine immune responses, it can be a useful framework for understanding immune responses dominated by either cell-mediated (T_H_1) or antibody-mediated (T_H_2) immunity [83]. In reality, the T cell response of cattle to various pathogens is heterogenous, although a predominant T_H_1 or T_H_2 response can and does occur [83]. In general, T_H_1 cells secrete cytokines such as IL-2, IFN-γ, and TNF-α, while T_H_2 cells produce cytokines such as IL-4, IL-5, IL-10 and IL-13 [84]. Clearance of intracellular pathogens, such as *M. bovis*, requires a T_H_1-biased response, characterized by infiltrates of inflammatory cells, increased phagocytosis, secretion of IL-2, IFN-γ and TNF-α, and generation of reactive oxygen and nitrogen species, which results in intracellular killing of pathogens [84]. In contrast, T_H_2-biased responses are characterized by secretion of IL-4 and IL-13, B cell proliferation, antibody production and cells or other factors which downregulate the inflammatory response. [84] These responses are not mutually exclusive and in the case of tuberculosis, the success or failure of an individual granuloma in controlling bacillary proliferation depends on the relative balance between T_H_1 and T_H_2 responses. As a result, it is not surprising that there are functional differences between granuloma stages and several bTB studies have examined cell composition and cytokine expression of the various stages. However, differences in routes of inoculation, inoculum dosages, animal age, animal breed, and methods used to examine cell surface markers, cytokine mRNA or cytokine proteins make comparisons between studies difficult [25,26,40,79,85,86,87,88,89,90] (Table 1).

Nevertheless, it is well accepted that an effective immune response to *M. bovis* relies on a T_H_1 cell-mediated response, controlled by cytokines released from antigen-specific T cells. IFN-γ is a crucial T_H_1-related cytokine necessary for granuloma development, macrophage activation and cytokine/chemokine induction [27,94]. Several studies have examined IFN-γ expression in bovine tuberculoid granulomas through qPCR, IHC or ISH, however, no clear pattern has emerged. Some studies have shown no difference in IFN-γ expression between granulomas of different stages, while others show differences between stages but without apparent increasing or decreasing trends [32,34,72,88]. One study reported an increasing trend of IFN-γ expression from stage I through stage IV granulomas [26]. A separate study reported no change in IFN-γ expression within granulomas examined at 5 weeks and 19 weeks post-infection, however, granulomas were not staged [87]. 

TNF-α produced by infected macrophages and T cells is key to both granuloma formation and maintenance [4,95], however, excessive TNF-α can lead to overt tissue damage. Both protein and mRNA of the proinflammatory cytokine TNF-α have been seen at variable levels in all granuloma stages [34,40,72,88], with a single study demonstrating an increasing trend from stage I through stage IV granulomas [26]. IL-10 is an important anti-inflammatory cytokine controlling innate and adaptive immune responses to *M. tb* complex organisms [96]. Its principal function is to deactivate macrophages, resulting in diminished T_H_1 cytokine production, as well as decreased reactive oxygen and nitrogen species production [97]. Studies in both experimentally and naturally infected cattle using ISH and IHC, demonstrated a positive correlation between granuloma stage and IL-10 mRNA expression, with the strongest immunoreactivity for IL-10 protein in stage IV granulomas [34,88]. In contrast, a study using ISH and experimentally infected cattle, examined 150 days after infection showed no significant difference in IL-10 expression between stage I granulomas and stage IV granulomas [72]. Comparison of IL-10 expression over time to individual granuloma bacterial burdens demonstrated that early in infection, when bacterial burdens and overall inflammation are high, IL-10 mRNA production is stimulated, while at later time points when bacterial burdens are low, IL-10 mRNA expression is also lower [34]. In humans, elevated levels of IL-10 expression have been associated with more active disease and more severe clinical signs [98,99].

In addition to T_H_1 and T_H_2 related cells, another distinct T helper cell lineage, T_H_17 cells produce, among other things, IL-17A. IL-17A has been associated with initiating both protective and harmful inflammatory responses to *M. bovis* [100]. In experimentally infected cattle, those with macroscopic lesions at days 15, 60 and 90 days after infection also had elevated levels of IL-17 in purified protein derivative of *M. bovis* (PPDB)-stimulated peripheral blood mononuclear cells (PBMC). A separate study observed increased IL-17 expression in stimulated PBMCs from BCG-vaccinated cattle, and further demonstrated that IL-17 may serve as a potential marker of vaccine efficacy [101]. Through IHC or ISH, several studies have examined IL-17A expression within granulomas. IL-17A mRNA expression is upregulated in granulomas of all stages with one study showing increased expression associated with stages I/II compared to stages III/IV pulmonary granulomas [24,40,72,79,102]. Moreover, increased IL-17 mRNA expression was greater in lung granulomas where increased neutrophilic infiltrates were noted compared to those with few neutrophils [79]. Additionally, upregulated in early stages of granuloma development are CXCL9 and CXCL10 (also knowns at interferon gamma-induced protein 10 (IP-10)), both potent chemokines for lymphocytes and macrophages. 

Experimental infection studies show that expression of key cytokines may be dependent on infectious dose. Although the level of pathology was similar between calves receiving an infectious dose of 1 colony-forming unit (CFU) compared to 1000 CFU, expression of IFN-γ, TNF-α, IL-4 and IL-10 increased with increasing dose in pulmonary lymph nodes 26 weeks after intrabronchial inoculation [86]. Most studies have evaluated bovine tuberculoid granuloma bacterial burdens using acid-fast staining techniques and found an increase in AFB in late-stage granulomas compared to early-stage granulomas. Importantly, the sensitivity of enumeration of bacterial burdens through microscopic examination is low, as AFB cannot be consistently found when tissue concentrations fall below 10^4^ mycobacteria/g [103]. Using quantitative bacteriological culture instead of acid-fast staining, a recent study of bacterial burdens of individual granulomas noted no correlation between granuloma stage and bacterial burden [34]. Of note, most studies also report that in early-stage granulomas AFB are present intracellularly in macrophages and MGC, while in late-stage granulomas most AFB are extracellular and free within the necrotic caseum [25,40,88]. In spite of their adaptation to intracellular growth and survival, bacilli readily replicate extracellularly in a mix of necrotic cell debris [4]. 

## 8. Necrosis and Distinct Spatial Arrangement

Granuloma architecture, composition, function and maintenance have a profound influence on the survival of mycobacteria within this niche [95]. The delayed-type hypersensitivity (DTH) response induced by infection with tuberculous mycobacteria results in necrosis within the granuloma. Lesion necrosis represents irreversible tissue damage. The gross morphological hallmark of this is the presence of semi-solid, cheese-like material termed caseous necrosis (Figure 5) [104]. 

Histologically, caseous necrosis has been described as a distinct form of coagulative necrosis and appears as an eosinophilic amorphous mass on slides stained with H&E [48]. In human tuberculoid granulomas, caseation is associated with a type of programmed cell death known as apoptosis, specifically the apoptosis of CD68^+^ macrophages, CD4^+^, CD8^+^ and CD45RO^+^ T cells [104,105]. 

The granuloma is the host’s attempt to both isolate and kill mycobacterial pathogens. In the process, however, inflammation and necrosis caused by the host response is often more detrimental than the pathogen itself. As granulomas develop and time after infection increases, there can be a conversion of the necrotic caseum from semisolid to liquid, likely the result of the action of various proteases and nucleases [106]. Expansive areas of necrosis may invade bronchi or bronchioles leaving cavities within lung parenchyma. This softened necrotic material, which may contain 10^7^–10^9^ CFU/mL, is coughed up and expelled [55]. In humans, cavity formation is one of the most important manifestations of tuberculosis in terms of transmission [107]. Cavitation is not a common manifestation of bTB today [107]; although in 1940, examination of 520 tuberculous cattle from the US and Canada revealed cavitation in one or both lung lobes in 131 (25%) of the cases [108]. Today, tuberculous cattle are generally identified earlier in the course of disease, consequently, severe cases are less common. This early detection may account for the lack of cavitary lesions today [109,110]. 

Granuloma enlargement is primarily due to an increase in central necrosis rather than an increase in cellular infiltrate [25,111]. In early granuloma stages, lymphocytes are admixed with macrophages, but as central necrosis increases and the granuloma enlarges, a distinct cellular arrangement can be appreciated. The area of central necrosis is surrounded by macrophages, epithelioid macrophages and MGCs, all of which may be positive for the surface transmembrane glycoprotein, CD68 [38]. As the distance from the necrotic core increases, the number of macrophages decreases while the number of lymphocytes increases, creating from outside in, a lymphocyte rich zone surrounding a macrophage rich zone, which surrounds the necrotic core [25,27,40]. Changes in spatial arrangement of cellular infiltrates may be associated with functional changes as well. In early lesions, when lymphocytes and macrophages are admixed, T cells, macrophages and MGCs are closely associated, facilitating cell-to-cell signaling critical for regulation of key cytokines, chemokines and enzymes [25,27,112,113]. As the granuloma expands, spatial arrangement is modified such that cell-to-cell apposition and cellular interactions are diminished [25]. 

In cattle, later stages III/IV are associated with increased peripheral fibrosis, which eventually encapsulates the granuloma within bands of collagenous connective tissue. It is believed this is due in part to abundant expression of TGF-β by macrophages, epithelioid macrophages and MGCs [114]. Coincident with increased fibrosis are increased levels of both type I procollagen and TGF-β [25,27,32,34]. Among the many effects of TGF-β, it increases extracellular matrix deposition needed for the granuloma framework and fibrosis in general [88]. In addition to fibroblasts, myofibroblasts have been described in mycobacteria-induced granulomas in humans. 

Healing, also characterized as fibrous organization has been described in the non-human primate and guinea pig models of human tuberculosis, specifically after drug therapy [115,116]. Such lesions have been characterized as extensive peripheral fibrosis with a central epithelioid macrophage population, organized areas of interstitial fibrosis, or foci of fibrosis and calcification [115,116]. In an examination of over 1500 human cases, between 1940 to 1944, Canetti described healing of lesions by the presence of calcification, fibrosis or ossification [117]. Although, in humans and models of human tuberculosis fibrosis and dystrophic calcification have been described as processes associated with healing [118], healing granulomas or fibrous organizing granulomas have not been described as such in cattle, possibly due to the lack of use of chemotherapeutic agents in bovine bTB. 

## 9. Mineralization

Mineralization (calcification) of necrotic material is termed dystrophic calcification. This pathological process is associated with intra-and extra-cellular deposition of mixed calcium salts at sites of necrosis. Calcification within necrotic granulomas is one of the most common examples of dystrophic calcification [119]. This can happen in any tissue, although in bTB lungs and lymph nodes are the most common. Calcified material appears dark blue to purple on H&E-stained sections, however, von Kossa staining better reveals the full extent of calcification. Dystrophic calcification is a local, organized process with deposition of crystalline hydroxyapatite calcium salts [120] and by definition, serum levels of calcium and phosphorus are normal. The cell type(s) and signal(s) responsible for mineral formation are unclear, and in both man and animals, the presence of calcified granulomas is of unknown significance. In cattle, some interpret the presence of calcification as a sign of chronicity, however, in experimental infections mineralization has been documented as early as 35 to 60 days after infection [33,82,121].

## 10. Conclusions

The bovine tuberculoid granuloma is a dynamic physical and functional structure that directly reflects the host: pathogen interface [122]. It is here that interactions between the host and pathogen determine bacterial killing or bacterial replication, disease confinement or disease dissemination. As such, understanding these interactions at the granuloma level is essential for development of improved diagnostic assays or effective vaccines. Some of the most significant advances in our understanding of tuberculosis pathogenesis have been elucidated using the cynomolgus macaque model of human tuberculosis [123]. Elegant research using this model has shown dynamic granulomas [122,124,125], changing in response to evolving interactions between bacteria and various host cell types [44,111,126,127]. This has been made possible through examination of individual granulomas using technology such as serial 2-deoxy-2-[^18^F]-fluoro-D-glucose (FDG) positron emission tomography (PET) with computed tomography (CT) imaging [122,128], which provides an in vivo assessment of individual granuloma activity. This FDG PET-CT data can be combined with data obtained postmortem, such as individual granuloma histopathology, bacterial burden, cytokine response, and gene expression to follow the course of infection within each granuloma. It is then possible to correlate bacterial burdens to a host of other readouts and determine the characteristics of granulomas that successfully control *M. tuberculosis* infection. In bTB, more work at the individual granuloma level will be needed to advance our understanding of bovine tuberculoid granuloma development, maintenance and bacterial killing capacity.

## Figures and Tables

**Figure 1 pathogens-11-00061-f001:**
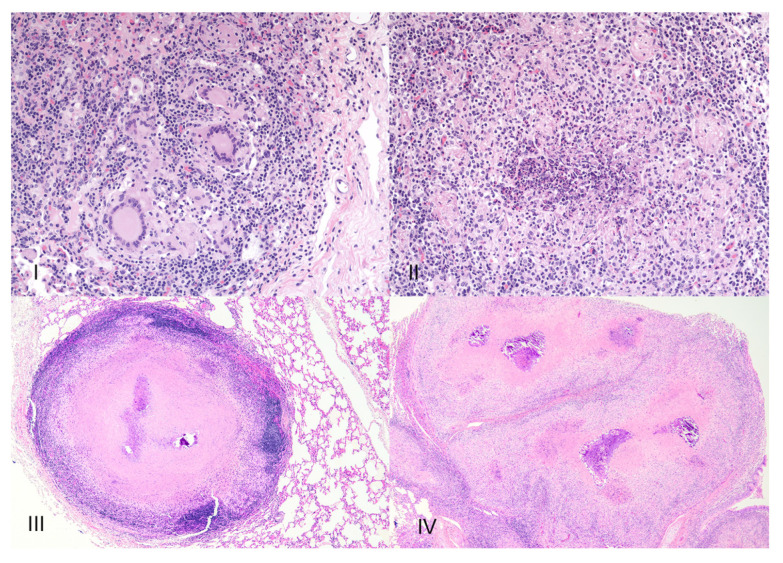
Four stages of bovine tuberculoid granulomas. Stage I granulomas contain infiltrates of epithelioid macrophages, lymphocytes and multinucleated giant cells. Stage II granulomas resemble stage I but have variable degrees of central necrosis. Stage III granulomas are composed of a central necrotic core, surrounded by a zone of macrophages admixed with lymphocytes and multinucleated giant cells. Portions of the necrotic core may be mineralized. Encapsulation of the necrotic core and cellular infiltrate by a complete fibrous capsule distinguishes this stage from earlier stages. Stage IV granulomas represent a coalescence of stage III granulomas with multiple necrotic and partially mineralized centers, surrounded and subdivided by thick fibrous bands.

**Figure 2 pathogens-11-00061-f002:**
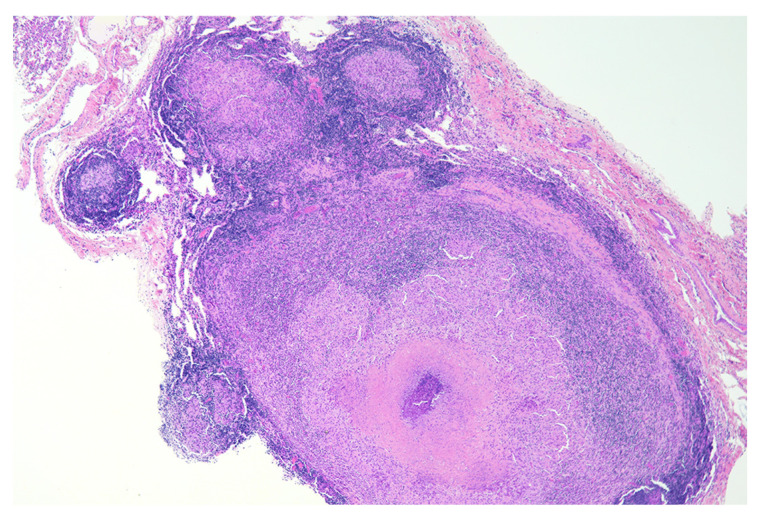
Section of lung from a calf experimentally infected with *M. bovis* via aerosol. Multiple satellite stage I granulomas are closely associated with a larger stage III granuloma. H&E.

**Figure 3 pathogens-11-00061-f003:**
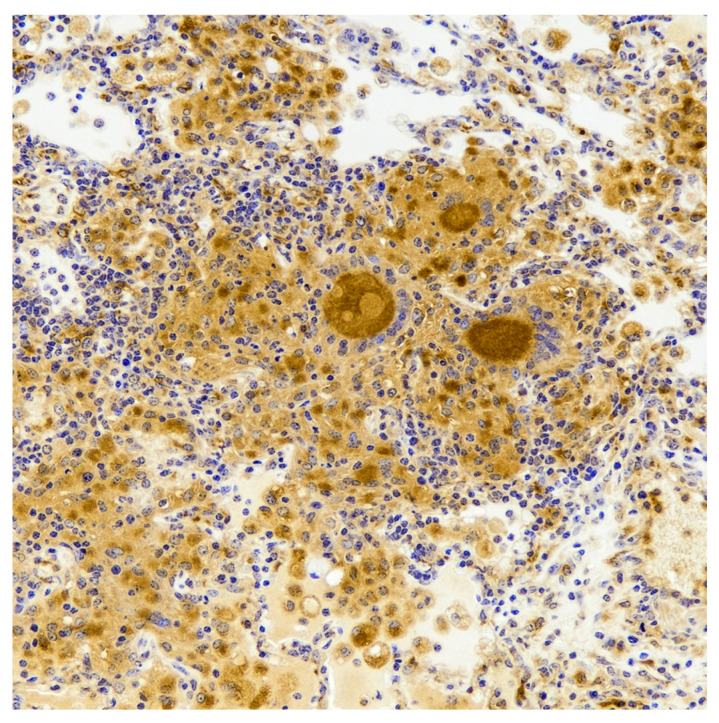
Section of lung from a calf experimentally infected with *M. bovis* via aerosol and examined 15 days later. Alveoli contain numerous macrophages and MGC, which stain positive for the macrophage marker CD68. IHC.

**Figure 4 pathogens-11-00061-f004:**
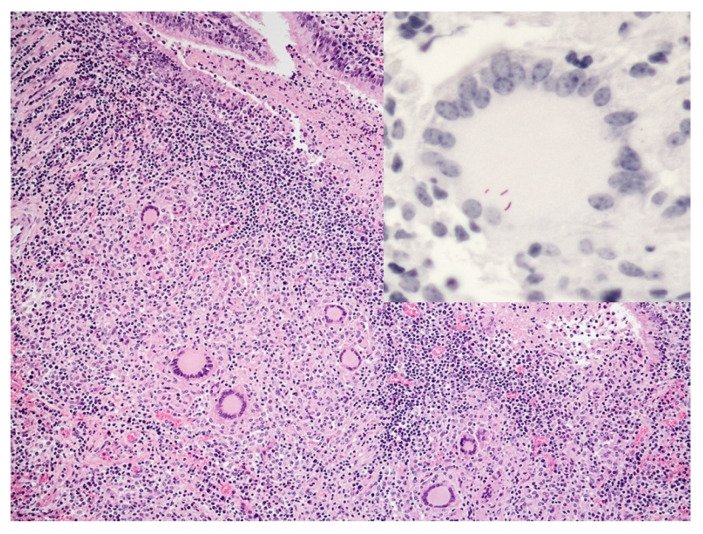
Section of lung from a calf experimentally infected with *M. bovis* via aerosol and examined 30 days later. Tuberculoid granuloma contains numerous multinucleated giant cells. H&E. Inset: multinucleated giant cell containing numerous magenta-colored acid-fast bacilli. Ziehl–Neelsen acid-fast stain.

**Figure 5 pathogens-11-00061-f005:**
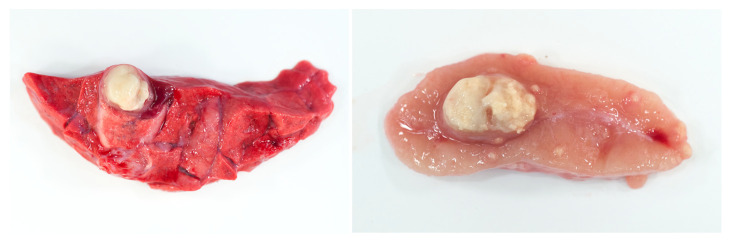
Samples of lung and lymph node contain central zones of caseous necrosis surrounded by pale thin fibrous capsules.

**Table 1 pathogens-11-00061-t001:** Summary of studies examining bovine tuberculoid granulomas.

Route of Infection	Dose (CFU)	Days after Infection	Age of Animal	Histopathology Description	Granuloma Staging	Cellular Identification	Cytokine Analysis	Reference
Natural	Unknown	Unknown	variable	Yes	Yes	No	No	[91]
Natural	Unknown	Unknown	18–30 months	Yes	Yes	NA	IHC	[88]
Natural	Unknown	Unknown	2.5–9 years	Yes	Yes	Yes	IHC	[85]
Natural	Unknown	Unknown	49–91 months	Yes	No	NA	IHC	[45]
Intratracheal	5 × 10^3^	28 weeks	15–16 months	Yes	Yes		qPCR	[79]
Intratracheal	70–6.2 × 10^4^	29 weeks	5 months	Yes	Yes	Yes	IHC/ISH	[25]
Intratracheal	10^4^	5–12 weeks	8 months	Yes	Yes	Yes	No	[37]
Intranasal	1.3 × 10^4^	12 weeks	3 months	No	No	NA	qPCR	[92]
Endotracheal	2 × 10^3^	13 weeks	5–7 months	Yes	Yes	Yes	IHC	[26]
Endobronchial	2 × 10^3^	13 weeks	12 months	Yes	Yes	Yes	IHC/ISH	[40]
Aerosol	10^3^–10^5^	21 weeks	4 months	Yes	No	No	No	[93]
Aerosol	10^4^	21 weeks	3 months	Yes	Yes	No	ISH	[24]
Aerosol	10^4^	12 weeks	6 months	Yes	Yes	Yes	ISH	[12]
Aerosol	10^4^	21 weeks	6 months	Yes	Yes	No	ISH	[72]
Aerosol	10^4^	30, 90, 180, 270 days	9 months	Yes	Yes	No	ISH	[34]

CFU = colony-forming unit; NA = not applicable; IHC = immunohistochemistry; ISH = in situ hybridization; q = quantitative PCR.

## Data Availability

Not applicable.

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
