# Peer review of "The Bovine Tuberculoid Granuloma"

_pathogens, 2022, doi:10.3390/pathogens11010061_

Round 1
Reviewer 1 Report
This is a very nicely written review on the bovine tuberculoid granuloma. The manuscript is comprehensively described, the references are adequates and up to date.
I really enjoyed reading the document and don't have much to say.
The description of granuloma staging and early lesions (pre-stade 1) is very precise.
To better follow the organization of the manuscript, I can propose to change the title of the section 7. "cellular response", which is bit vague, for instance when the authors described the macrophages and MGC, this is also cellular response to me). A change for "Lymphocytes response" or "cellular mediated adaptive immunity"...
My only regret is regarding the section 8. on neutrophils. Even though not much is known in cattle, these cells are very important for the granuloma formation in human TB. Maybe the authors can add few sentences to emphasize a bit more their key role in (human) TB : neutrophils subsets with distinct functions (proinflammatory versus regulatory), role in granuloma formation etc.
Author Response
We appreciate the reviewer's suggestions on the organization of the sections on various cellular responses. We have reorganized the paper such that all the cellular responses are adjacent to each other. We feel this increases the readability of the paper.
The section on neutrophils has been expanded.
Reviewer 2 Report
The review on the bovine tuberculoid granuloma presented by Palmer et al is .
I only have a few concerns.
Italics on microorganism names are missing throughout the manuscript. These are in lines 123, 142, 155, 192, 263 and 340. I recommend going through the manuscript carefully as i may have missed some.
Figure 4. I'm not sure if it is my computer but the ZN staining is not clear. I'm not able to see AFB in the MNG cells. I suggest improving quality and/or adding arrows to help localization.
In the section dedicated to neutrophils on Page 12; this parragraph is loose, low cohesion in what is being described. In lines 454-455 when authors state that neutrophils form extracellular traps against Mbv BCG in vitro effectively killing them I miss a reference. It can be "Ladero-Auñon Front Immunol 2021;12:645304. doi:10.3389/fimmun.2021.645304.
Author Response
We apologize for the oversight concerning italics and organism names. We have made these changes in the text and figure legends.
The inset for Fig 4 has been changed and enlarged to clearly show the acid-fast bacteria.
The section on neutrophils has been expanded.
The mission reference, Ladero-Auñon Front Immunol 2021;12:645304. doi:10.3389/fimmun.2021.645304 has been added.
Reviewer 3 Report
The review on “The bovine tuberculoid granuloma” by Palmer, Kanipe and Boggiatto is a complete synthesis on the pathology of tuberculosis in cattle. I have only very few comments.
- The differences between naturally occurring and experimental mycobacterial infection should be specified
- The pattern and mechanisms of healing need a specific paragraph in addition to the one on mineralization/calcification. For example, the absence of concentric fibrosis observed in humans.
- The different patterns observed in the involved organs should be discussed
Author Response
The authors appreciate the reviewer's comments. The manuscript is meant to focus on granulomas due to M. bovis infection in cattle. Comparisons with granulomas in other species, primarily humans or models of human tuberculosis are made when necessary, such as when mechanisms of pathogenesis have not been fully elucidated in cattle.
When studies involving naturally infected cattle are referenced, it is clearly stated. Using natural infection studies to characterize granulomas has the advantage of being natural in the sense of dose and route of infection, but the duration of infection is unknown. Dose and route of infection are controlled in experimental settings and duration of infection is known. This is critical in describing a chronology of events in granuloma development. In the last paragraph before table 1 the authors state the difficulty in comparing natural and experimental infection studies. In table 1 we have listed the nature of infection, natural or experimental.
In the sections on necrosis and mineralization the authors have added text discussing the morphology of "healed" lesions in models of human tuberculosis and natural cases of human tuberculosis. The authors are not aware of literature describing lesions as "healed" or "healing" in bovine tuberculosis.